# Electromechanical Properties of PVDF-Based Polymers Reinforced with Nanocarbonaceous Fillers for Pressure Sensing Applications

**DOI:** 10.3390/ma12213545

**Published:** 2019-10-29

**Authors:** Javier Vicente, P. Costa, S. Lanceros-Mendez, Jose Manuel Abete, Aitzol Iturrospe

**Affiliations:** 1Electronics and Computing Department, Mondragon Unibertsitatea, 20500 Mondragon, Spain; jvicentet@mondragon.edu (J.V.); aiturrospe@mondragon.edu (A.I.); 2Center of Physics, University of Minho, 4710-057 Braga, Portugal; 3Institute for Polymers and Composites (IPC), University of Minho, 4800-058 Guimarães, Portugal; 4BCMaterials, Basque Center for Materials, Applications and Nanostructures, UPV/EHU Science Park, 48940 Leioa, Spain; senentxu.lanceros@bcmaterials.net; 5IKERBASQUE, Basque Foundation for Science, 48013 Bilbao, Spain; 6Applied Mechanics Department, Mondragon Unibertsitatea, 20500 Mondragon, Spain; jmabete@mondragon.edu

**Keywords:** piezoresistivity, PVDF, nanocarbonaceous, electromechanical, pressure sensibility

## Abstract

Polymer-based composites reinforced with nanocarbonaceous materials can be tailored for functional applications. Poly(vinylidene fluoride) (PVDF) reinforced with carbon nanotubes (CNT) or graphene with different filler contents have been developed as potential piezoresistive materials. The mechanical properties of the nanocomposites depend on the PVDF matrix, filler type, and filler content. PVDF 6010 is a relatively more ductile material, whereas PVDF-HFP (hexafluropropylene) shows larger maximum strain near 300% strain for composites with CNT, 10 times higher than the pristine polymer. This behavior is similar for all composites reinforced with CNT. On the other hand, reduced graphene oxide (rGO)/PVDF composites decrease the maximum strain compared to neat PVDF. It is shown that the use of different PVDF copolymers does not influence the electrical properties of the composites. On the other hand, CNT as filler leads to composites with percolation threshold around 0.5 wt.%, whereas rGO nanocomposites show percolation threshold at ≈ 2 wt.%. Both nanocomposites present excellent linearity between applied pressure and resistance variation, with pressure sensibility (PS) decreasing with applied pressure, from PS ≈ 1.1 to 0.2 MPa^−1^. A proof of concept demonstration is presented, showing the suitability of the materials for industrial pressure sensing applications.

## 1. Introduction

Polymer-based nanocomposites are attracting large attention in recent years both in the scientific and industrial areas. Through the inclusion of fillers, a wide range of polymer properties can be enhanced, such as mechanical [1], electrical [2], and thermal properties [3], among others. Thus, polymer composites can be tailored for specific applications. Moreover, it is possible to add new features to the polymers through the inclusion of nanoparticles, functionalizing them and enabling sensing of different parameters such as mechanical [4], temperature or humidity [5], among others physical properties.

Hence, their application as sensors materials are being widely studied due to their simple manufacturing and integration into devices [6,7]. Some of the most used fillers for strain sensing functionalization of polymers are carbon nanoallotropes, such as carbon black (CB) [8], graphene (G) and its oxidized forms (graphene oxide (GO) and reduced GO (rGO) [8,9] and carbon nanotubes (CNT) [7,10]. With the inclusion of conductive nanoparticles, polymer nanocomposites increase their piezoresistive response, i.e., under a mechanical solicitation their resistance changes linearly with applied strength [11]. Different fillers influence the electrical and functional properties, but also the mechanical properties of the composite [11,12]. The percolation threshold of the polymer composites depends on the filler, matrix, and processing method, among other parameters [13]. Lower percolation thresholds around 0.1 to 0.4 vol.% can be found in polymer composites with graphene or carbon nanotubes as fillers [13]. The aspect ratio of the nanofillers as well as their intrinsic properties strongly influence the electrical and mechanical properties of the composite and, therefore, its functional response [12]. In this way, graphene and CNT are compared as reinforcement fillers due to their different intrinsic properties in order to find suitable nanofillers for specific functional devices. These materials are typically tailored to optimize their mechanical properties and piezoresistive sensibility, achieving gauge factor values up to 175 [14], two orders of magnitude higher than traditional strain gauges, and strains larger than 50 % [15]. The percentage of fillers used for optimizing functional response also vary widely [16], depending on the matrix, filler, processing method or even application. However, the larger piezoresistive sensibility in polymer composites is observed near the percolation threshold [17].

The percolation threshold is the range in which the materials undergo a transition from nearly insulating to conductive, changing several orders of magnitude their electrical conductivity for small filler content variations [16]. The percentage at which this phenomenon occurs varies widely depending on fillers, matrices, and processing methods [18,19]. The percolation threshold is reported to occur when the first conductive paths spanning all the nanocomposite are formed due to the proximity of the conductive fillers embedded in the isolating matrix. This phenomenon has been explained by different theoretical models [19,20,21].

The selection of the polymer matrix used for the development of a functional sensor depends on the stimulus needed for sensor response. Flexibility or stretchability, force, and environmental conditions influence the host polymer to use in view the overall properties to select.

Soft polymer matrices such as natural rubbers or thermoplastic elastomers are reported to provide the nanocomposite high strain capability from low to large strains [15], and a wide range of sensitivity in functional response [22]. For applications in which mechanical solicitations can compromise the structural integrity of the material or for large force applications, stiffer matrices are employed. Among the most used thermoplastic polymers for force and deformation sensor development, literature reports on polypropylene (PP) [23], poly(vinylidene fluorine) (PVDF) [23,24] and poly-ether-ether-ketone (PEEK) [25], among others [25,26], though thermosetting such as polyepoxides [27] and elastomers such as thermoplastic polyurethane (TPU) [8,9], triblock styrene-butadiene-styrene (SBS) [15] or styrene-ethylene/butylene-styrene (SEBS) [23] have been also used.

PVDF and its copolymers are excellent materials for functional applications, such as sensors, actuators, energy harvesting and biomaterials in the biomedical field [28]. PVDF shows excellent electroactive properties, being used as host polymer for large number of applications [28]. PVDF is semi-crystalline material with five distinct crystalline phases, the most investigated and used for applications being the non-polar α-phase and the polar β-phase [11,27]. Furthermore, PVDF presents excellent mechanical and chemical properties, weather resistance, and outstanding properties associated with their polar crystalline forms [29]. In this way, functional composite applications is an interesting material with large potential for force sensor, due to its mechanical and chemical resistance properties [12,27].

Polymer nanocomposites can be manufactured using laboratory and industrial techniques. At the laboratory level, they can be processed by solvent casting [30], for example, whereas at industrial level typical processing methods include hot pressing [31], extrusion [32], or injection [33]. Through the different fabrication processes, the overall properties of the PVDF composites can be tuned, including mechanical and electrical properties, with large influences on the functional performance of the composite. In particular, thin-film materials attract increasing attention based on their simple integration in to devices [33,34,35,36]. Hence, the integration of these films in components or processes enables an inexpensive sensor conformation.

In this work, a comprehensive study of PVDF-based materials for force compression sensing is presented with excellent performance and linearity [37,38]. Different PVDF based polymers (PVDF-HFP (poly(vinylidene fluoride-co-hexafluropropylene)), PVDF 6010 and PVDF 5130) and nanocarbonaceous fillers (carbon nanotubes and reduced graphene oxide) were used to tailor composites in view their mechanical, electrical and electromechanical properties. Further, proof of concept application is presented, submitting the sensor to different pressures. In order to develop a functional material with specific responses to the applications, the focus on the different properties of the polymer matrices and fillers (type and content) will be evaluated to tailor the overall properties of the composite to work as piezoresistive sensible material under mechanical compression. Host matrices were selected from 400 MPa to 2.5 GPa of tensile modulus, as detailed in experimental part and CNT and rGO have been selected as functional fillers based on their different dimensions and intrinsic properties.

## 2. Experimental

### 2.1. Materials

The selected polymers were poly(vinylidene fluorine) with reference 5130, 6010, and PVDF-HFP, all supplied by Solvay. PVDF 5130 is characterized by ultra-high viscosity with excellent adhesion, a density of 1.75 g/cm^3^ and a tensile modulus between 1 to 1.5 GPa. PVDF 6010 is a homopolymer with medium viscosity, density between 1.75 to 1.8 g/cm^3^ and a tensile modulus between 1.7 to 2.5 GPa. Poly(vinylidene fluoride-co-hexafluropropylene) (PVDF-HFP), with reference Solef 21,216 and VDF/HFP mole ratio of 88/12, shows a density of 1.78 g/cm^3^ and a tensile modulus between 400 to 600 MPa.

The solvent used to disperse the nanofillers and dissolve the PVDF was *N*,*N*′-dimethylpropyleneurea (DMPU) and was purchased from LaborSpirit.

Multi-walled carbon nanotubes were supplied by Nanocyl with reference NC7000, showing an average length of 1.5 μm, an outer mean diameter of 9.5 nm and 90% purity. Reduced graphene oxide was obtained from The Graphene Box (Spain) with >99% of purity, 1–5 µm of length and 1–2 layers.

### 2.2. Sample Preparation

Carbon nanofillers were dispersed in DMPU within an ultrasonic bath (ATU, Model ATM40-3LCD) for an average time of 4 h, assuring a correct de-agglomeration and homogeneous dispersion of the fillers in the solvent. Then, PVDF (5130, 6010, or HFP) was added to the filler/solvent solution and completely dissolved through magnetic stirring for approximately 3 h at 30 °C. It is to notice that this processing method has demonstrated to ensure good filler dispersion for both graphene [12] and CNT [15]. Then, the solution was spread on a clean glass substrate by doctor blade method with a 100 µm blade thickness. Finally, films were melted in an oven at 210 °C for 25 min, promoting the crystallization of the PVDF in the α-phase and achieving complete solvent evaporation [39]. The thicknesses of the films after complete evaporation of the DMPU solvent ranges from 20 to 60 µm.

Films with the different polymer matrices and carbonaceous filler percentages were prepared, as indicated in Table 1.

### 2.3. Sample Characterization

Fourier-transform infrared (FTIR) spectroscopy analysis was carried out in transmission mode at room temperature from 4000 cm^−1^ to 600 cm^−1^ with a resolution of 4 cm^−1^ employing a Jasco FT/IR-4100 spectrometer with a TGS detector.

Differential scanning calorimetry (DSC) tests were performed with a Netzsch DSC 200F3 Maia set up. Samples were placed into Al pan crucibles, stabilized at 30 °C and then subjected to a 20 °C.min^−1^ heating rate up to 200 °C under nitrogen atmosphere.

Electrical conductivity was obtained after I–V measurements. Samples were submitted to a voltage sweep from −10 to +10 V and the current was measured with a Keithley 6430 SourceMeter. Gold electrodes were previously deposited in both sides of the films by magnetron sputtering with a Polaron SC502 sputter coater. The electrical conductivity was obtained after Equation (1) considering sample thickness and electrode area:(1)$$\sigma =\rho^{-1}={\left(R\frac{A}{L}\right)}^{-1}$$
where *R* is the electrical resistance, *A* the electrode area and *L* the sample thickness. Electrical conductivity tests were performed in two sample points and repeated three times each.

The mechanical characterization of the nanocomposites was performed by tensile tests up to failure with a Shimadzu AG-IS universal testing machine with a 50 N load cell, repeated for 5 measures for each sample.

For the tensile tests, samples of 30 × 10 mm of the area and 20 to 60 μm of thickness were cut from the fabricated films, placed in the universal testing machine through a clamping system leaving an approximate interspace between clamps of 10 mm to assure a proper grip of the sample. The measurements were performed at speed of 1 mm/min and force and test time were recorded at a 100 ms rate. Engineering stress and strain (Equations (2) and (3)) were obtained according to:(2)$$\sigma =\frac{F}{A}$$
(3)$$\varepsilon =\frac{\Delta L}{L_{0}}$$
where *F* is the tensile force, *A* the transversal area of the sample considering the thickness and width of the sample, Δ*L* the recorded displacement and *L*_0_ the initial interspace between clamps. Tensile tests were repeated for three times, ensuring that measurements were consistent within materials. The samples were considered as macroscopically homogeneous materials, without considering non-local effects [40] due to the low filler content.

Electromechanical tests were performed under cyclic compression on samples with initial minimum pre-load of 10 N corresponding to 0.09 MPa to forces of 50, 100, 200, and 400 N, corresponding to 0.43, 0.86, 1.72, and 3.45 MPa, respectively, considering the contact area of the compression tests. The piezoresistive tests were performed in a Shimadzu AG-IS universal testing machine with a load cell of 500 N at speed of 0.5 and 1 mm/min, for 10 and 50 cycles and replicated twice. Force and displacement were recorded with a 500 ms time span.

The acquisition of the electrical resistance between the compression pieces was performed with an Agilent 344401A multimeter. Piezoresistive tests were repeated twice.

The quantitative evaluation of the pressure sensitivity (*PS* in Equation (4)) was performed according to:(4)PS= ΔRR0P
where Δ*R* represents the resistance, variation measured during the test and *R*_0_ the initial resistance of the sample under the minimum pre-load and *P* is the pressure on the sample.

## 3. Results and Discussion

### 3.1. Chemical and Thermal Characterization

Figure 1 presents the FTIR spectra of the different neat semi-crystalline PVDF polymer (Figure 1A) and the corresponding composites reinforced with CNT and rGO (Figure 1B). The presence of α-phase in the samples is confirmed by the corresponding bands at 614, 763, 795 and 975 cm^−1^ [28], whereas vibration bands corresponding to the β, γ, and δ [28] phases are detected neither in the neat polymers or the corresponding composites. In fact, the α-PVDF phase is the one obtained when the polymer is crystallized from the melt [28]. No significative variations in the FTIR spectra were detected among the different composites.

Literature reports that some nanoparticles such as carbon nanotubes can influence the crystallization phase of PVDF [41]. The neat conductive fillers CNT and rGO, without surface modification, do not influence the crystallization phase of the polymers as observed in the FTIR spectra (Figure 1B) for the PVDF composite materials, as reported previously [12]. Neat nanocarbonaceous materials do not act as nucleating agents [41] in PVDF, leading just to α-phase crystallization independently of filler type and content.

Thermal properties were evaluated by DSC thermograms for neat PVDF 5130, 6010 and HFP (Figure 2A) and their corresponding nanocomposites with CNT or rGO with different filler content (Figure 2B). Higher variability on the melting temperature (*T_m_*) can be observed between the different PVDF matrixes, with PVDF-HFP presenting lower crystallization temperature, *T_c_* ≈ 132.3 °C, than PVDF5130 and 6010, with *T_m_* ≈ 158.2 and *T_m_* ≈ 168.9 °C, respectively. The thermograms of the neat polymers and the nanocomposites show a single peak corresponding to the melting of the α-phase of PVDF [42].

PVDF composites with CNT or rGO lead to crystallization temperatures slightly higher than the neat polymers, increasing less than 2 °C in all cases, demonstrating that the effect of the fillers in the melting behavior of the samples is quite small and independent of filler type and content, confirming the low interaction between fillers and polymers observed in the FTIR spectra (Figure 1), where the introduction of the fillers lead to no new chemical bonds or variations in the polymer bonds.

### 3.2. Electrical Characterization

The volume of electrical conductivity for the different composites is shown in Figure 3. Electrical conductivity increases with the inclusion of both nanofillers for all PVDF copolymers, showing a percolation threshold lower than 0.5 wt.% for CNT/PVDF composites and near 2 wt.% for rGO/PVDF5130 composites. The conductive network formed by the CNT is more effective than for rGO for similar filler contents into the PVDF matrix, the intrinsic conductivity of the CNT being higher than the one of the rGO nanofillers. Further, the aspect ratio of the CNT is larger than the one of the 2D materials, leading to lower percolation threshold composites. The conductivity of the neat polymers is in agreement with the manufacturer specifications and is similar among PVDF copolymers [11,42,43]. Among CNT nanocomposites, PVDF6010 shows the higher electrical conductivity for filler content between 0.25 and 0.5 wt.% when compared to PVDF-HFP and 5310 composites. The electrical conductivity for CNT/6010 is higher for 0.25 wt.% CNT (near 2 orders of magnitude) but for composite with 0.5 wt.% filler content the electrical conductivity is in the same order of magnitude for all polymer matrices. Their percolation threshold is thus around 0.5 wt.% CNT and the maximum conductivity is σ ≈ 5 × 10^−1^ (Ω.m)^−1^, as reported in literature for CNT/polymer materials [43,44]. It is to notice that percolation thresholds below 0.1 wt.% of CNT have been reported for PVDF matrix composites [44], which is not verified in the present work.

With respect to rGO/5130 nanocomposites, they show lower conductivity when compared to CNT at the same filler content, which is attributed to the lower aspect ratio and intrinsic conductivity of rGO when compared to CNT, leading to an increase of percolation threshold of the composite. The rGO/5130 up to 1 wt.% filler content shows similar electrical conductivity than neat PVDF5130 and the percolation threshold is ≈ 2 wt.% rGO, with an electrical conductivity of σ ≈ 1 × 10^−5^ (Ω.m)^−1^.

The intrinsic properties of CNT lead to low percolation thresholds in polymer-based composites. To tailor polymer-based nanomaterials with functional properties, low nanofiller content is typically required in order not to affect other properties of the polymer, such as thermal or mechanical. In terms of higher conductivity and lower threshold, CNT appears as more interesting nanocarbonaceous filler than rGO for conductive polymer nanocomposites.

### 3.3. Mechanical Measurements

Mechanical measurements were performed to evaluate the stress-strain response of the several PVDF polymers and composites, as a function of the filler content and type. As it is shown in Figure 4, CNT tends to reinforce the PVDF matrix, leading to a higher stiffness for the composites when compared to neat matrix [45,46,47,48]. However, at the higher CNT contents, maximum stress increases, while the maximum strain of the composites decreases. This effect is ascribed to the heterogeneity of the composites caused by filler agglomeration [49] that limits the mechanical strain for all PVDF matrices, PVDF-HFP, 6010, and 5130. The maximum stress of the CNT/PVDF samples, near the yielding of the PVDF and composites, increases with increasing CNT content. This means that the CNT effectively mechanically reinforces the composites, as presented in Table 2, and that the CNT agglomerates can act as mechanical interlocking between polymer chains and the filler [49]. In fact, it has been shown in different graphene/PVDF [12] and CNT/PVDF [15] composites with low filler content into the PVDF matrix, that their presence do not influence the spherulitic size and the kinetics of crystallization, the cross-section SEM images demonstrating a homogeneous dispersion of the nanocarbonaceous fillers, independently of the filler type and content [12]. Morphological analysis of the nanocarbonaceous/polymer composites have been intensively studied in literature [7,50,51], the present results being in agreement with the reported literature (data not shown).

Among the different polymers, PVDF6010 shows the highest yield strength (larger than 40 MPa) and PVDF-HFP the lowest yield stress (lower than 20 MPa), being inversely for strain at rupture, where PVDF-HFP shows the larger strain, near 70%, and the 6010 rupture is near 12%. At rupture, composites with CNT embedded in PVDF-HFP and 5130 present larger maximum strain, reaching 200% of strain for 025CNT/5130 and 300% for 025CNT/HFP samples. In all the composite samples, except the ones including rGO, the inclusion of CNT fillers yields to higher stiffness and elongation at break. From the behavior observed in the rGO/PVDF samples, it can be concluded that the matrix-filler wetting is considerably weaker than the PVDF-CNT one, leading to brittle fracture of the samples. In fact, literature has reported on the decreasing of the maximum strain with low graphene content as reinforcement for PVDF 1010 [12].

The initial modulus for neat PVDF is larger for 6010 near *E* ≈ 1 GPa being *E* ≈ 870 and *E* ≈ 350 MPa for 5310 and HFP, respectively. Reinforced with CNT, all PVDF matrices increase the initial modulus with increasing CNT content, excepting for the 1CNT/5310 sample. Similar behavior is found in rGO/5310 composites, where the initial modulus increases with filler content up to 1 wt.%, decreasing for samples with 2 wt.%.

The 05CNT/6010 sample shows an initial modulus near *E* ≈ 1.4 GPa. The larger initial modulus and yield stress of the PVDF 6010 composites are correlated with the percolation threshold, which depends on the filler dispersion, but also rigidity of the matrix [52], decreasing the threshold below 0.5 wt.%.

The strain at rupture for composites with PVDF 6010 is similar for the different CNT contents, increasing in the remaining PVDF composites, from 70 to 300% in HFP and 23 to 210% in 5130. PVDF5130 with rGO filler suffers a decrease of the strain at rupture from 23 to 4.5%, from neat polymer to the composite with 2 wt.% rGO, respectively. The yield stress and strain are comparable for both fillers (CNT and rGO) in PVDF 5130 materials. The load transfer efficiency and interfacial shear stress of the composites as a function of the dimensions of the fillers have been theoretically calculated [53]. The CNT and rGO fillers employed in the present work show average lengths of 1.5 μm and 1 to 5 μm for CNT and rGO, respectively. The CNT diameter is 9.5 nm and the rGO thickness is 1–2 layers that correspond to 2–3 nm [54]. Thickness and length of the employed nanocarbonaceous fillers are similar, being different the width of both materials, from 9.5 nm to 1–5 μm, respectively, for CNT and rGO. In this way, theoretical and experimental results are in agreement, the larger nanofillers leading to better reinforcement of the composites [55], as observed in the rGO/5130 composites, which show slightly higher initial modulus when compared to CNT/5130 composites.

Those results reflect not just the different mechanical characteristics of the polymers but also the different wettability between polymer and fillers, depending on macromolecule characteristics.

### 3.4. Electromechanical Measurements

PVDF composites with CNT or rGO as filler are excellent candidates for electromechanical sensors, leading to high sensibility composites [11,24]. Electromechanical compression tests (in Figure 5) were performed in the composites around the percolation threshold to evaluate the sensitivity and linearity of the different materials. The mechanical stress-strain cycles applied to the samples up to 3.45 MPa are shown in Figure 6. It can be observed a slight nonlinear response and that the mechanical hysteresis is very low for the 05CNT/5130 composites, being similar for the different nanocomposites.

Figure 6 shows 10 loading-unloading tests performed at a maximum load of 3.5 MPa (400 N of force) for the different composites with 0.5 wt.% CNT content and the 2rGO/5130 sample. Electromechanical tests show good linearity between electrical resistance variation and applied pressure for the different matrices and fillers, as a function of the pressure. Similar electromechanical linearity and cycling performance have been reported in [56,57] with different materials and experimental approaches.

The composites present larger piezoresistive sensibility near the percolation threshold [58]. The linear behavior between the applied pressure and electrical resistance variation is present in all composites, for loading and unloading cycles, as shown in Figure 6 and Figure 7A.

The pressure sensibility was evaluated in the different nanocarbonaceous/PVDF composites as a function of polymer and filler type and applied pressure. The highest sensitivity is obtained for 05CNT/HFP composite with PS ≈ 1.1 MPa^−1^, as shown in Figure 7. Further, as it is shown in Figure 7B that the piezoresistive sensitivity decreases with increasing applied pressure due to the compression of the filler network and, therefore, the filler-filler distance. This is in agreement with related CNT-based nanocomposites, showing the opposite behavior with respect to test performed under tensile electromechanical conditions [16,59]. Further, all composites present good linearity between electrical resistance variation and applied pressure during the compression cycles, with CNT/6010 and rGO/5130 composites showing the larger piezoresistive sensitivity. On the other hand, although showing good sensitivity and piezoresistive response, 05CNT/5130 composites show lower electrical stability (Figure 6C) when compared with the other composites.

### 3.5. Proof of Concept Application

The validation of the sensor in pressure sensing applications was performed using the 05CNT/6010 composite. The CNT nanocomposite was integrated into a developed test bench built to emulate the mechanical stiffness present in different mechanisms. With this use case, the capability of employing the developed materials to fabricate sensors which could be applied in industrial smart components is evaluated.

The test bench, shown in Figure 8, consists of a PMDC motor which through a torque coupler actuates in a threaded rod, producing a net displacement of the tip of the rod. This tip rests against a stainless-steel cantilever beam, which opposes the rod axial displacement. Thus, the PMDC motor produces a torque in the rod that is translated into an axial force which deflects the beam. The motor has a gearbox of 31:1. Considering that the thread pitch is 1 mm, the relationship between motor revolutions and rod tip displacement is 32.3 µm/rev.

The nanocomposite material was integrated between the beam and the threaded rod. The resistance change produced under the compression pressure was measured between the fixtures of the beam and the rod employing a Fluke 8845A multimeter and logged through PC. The PMDC motor (Maxon EC-4 pole 22 mm) was actuated by an EPOS2 controller.

Figure 8 shows the sensor readout when submitted to 8 cycles of 4 revolutions forward and back, as a representative performance of the piezoresistive sensor. The angular velocity was 750 rpm for the first 4 cycles and 1500 rpm for the latter four cycles.

Figure 8 shows that the sensor readout presents high repeatability between the cycles both at a high and low rotational speed. No significant drift is presented in the measured signal, confirming the suitability of the developed materials for sensor applications. The readout circuit shows electrical resistance signal of the composite following the pressure applied to the composite, as observable in Appendix A, and the electrical signal stabilizes after removing the pressure.

## 4. Conclusions

Different PVDF copolymers were reinforced with carbon nanotubes (CNT) and reduced graphene oxide (rGO) fillers to evaluate the performance of the materials for piezoresistive sensor applications.

FTIR analysis shows that PVDF crystallizes in the α-phase, independently of polymer type and filler type and content. Similarly, the mechanical tensile modulus of the matrix and reinforcement filler (CNT or rGO) do not influence filler dispersion for low filler contents. Thermal measurements show the melting temperature around 132, 158, and 169 °C for PVDF-HFP, 5310 and 6010, respectively. This temperature is just slightly affected by the inclusion of the fillers. With respect to the mechanical response, PVDF 6010 presents the highest initial modulus. Tensile tests demonstrate that the inclusion of fillers reinforces the polymer matrices, leading to higher stiffness, yield strength or elongation at break depending on the percentage used. Their percolation threshold is lower for the CNT nanocomposites when compared with the rGO ones, being the percolation threshold independent on the polymer matrix.

Pressure sensibility is larger for PVDF-HFP with 0.5 wt.% CNT for low applied pressures. For the largest pressure, the PS is similar for all materials. The electromechanical pressure sensibility of the materials as a function of pressure varies between 0.2 < *PS* < 1.1. The linearity between the electrical resistance variation and pressure is present in all composites. Finally, a proof of concept is presented showing the suitability of the materials for applications. Hence, PVDF/CNT and rGO based piezoresistive nanocomposites present suitable characteristics to work as embeddable, highly sensitive and cost-effective sensors in industrial pressure sensing applications.

## Figures and Tables

**Figure 1 materials-12-03545-f001:**
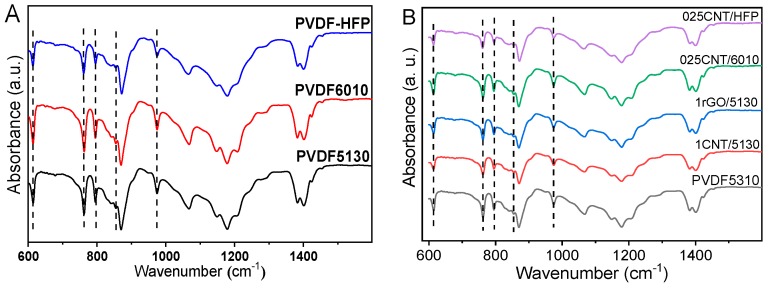
Fourier-transform infrared (FTIR) spectra of (**A**) neat polymers and (**B**) composites with carbon nanotubes (CNT) or reduced graphene oxide (rGO) nanofillers.

**Figure 2 materials-12-03545-f002:**
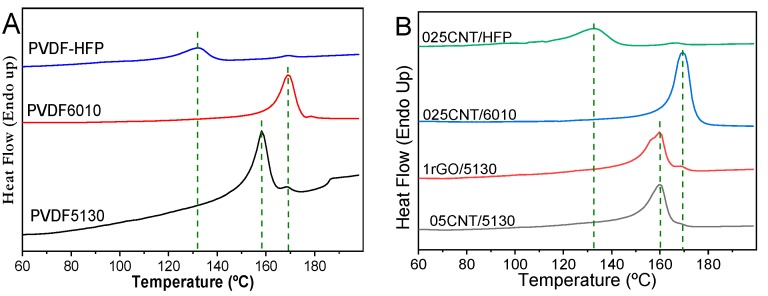
Differential scanning calorimetry (DSC) scans of (**A**) neat PVDF 5310, 6010 and HFP (hexafluropropylene) and (**B**) corresponding nanocomposites with CNT and rGO for different filler contents.

**Figure 3 materials-12-03545-f003:**
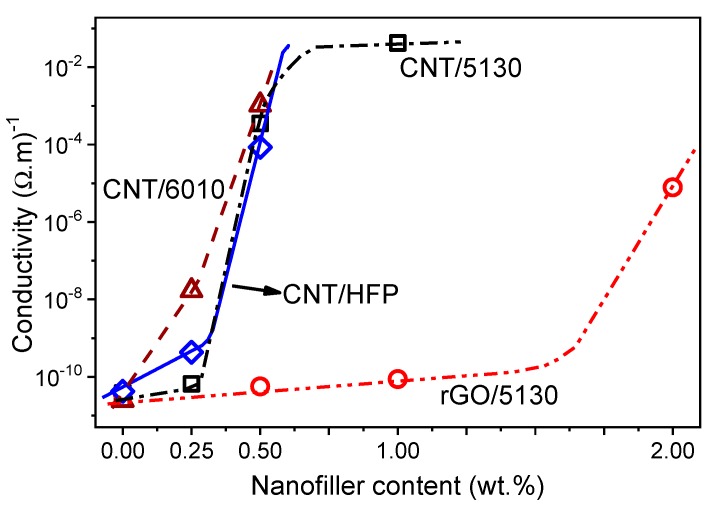
Electrical conductivity of the fabricated samples as a function of filler type and content. The lines are for guiding the eyes.

**Figure 4 materials-12-03545-f004:**
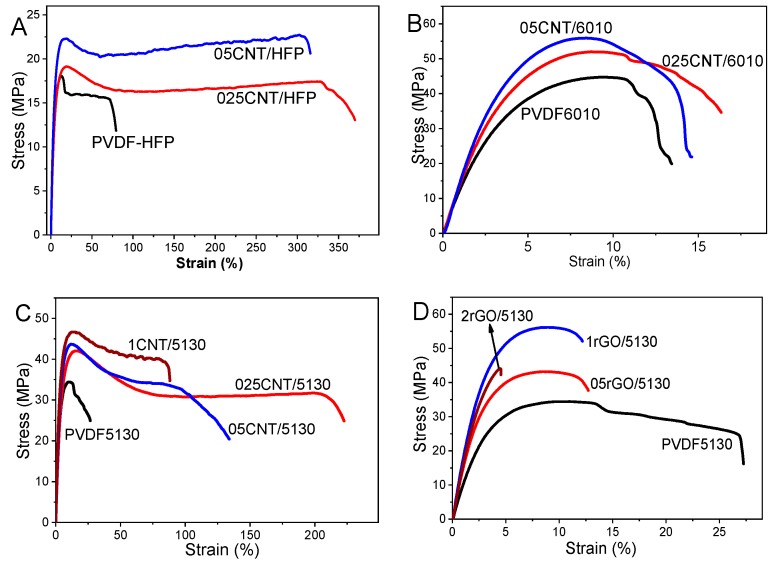
Stress-strain response for PVDF and the corresponding composites for (**A**) PVDF-HPF, (**B**) PVDF6010 and (**C**) PVDF5010 for different CNT contents. (**D**) PVDF5130 reinforced with rGO.

**Figure 5 materials-12-03545-f005:**
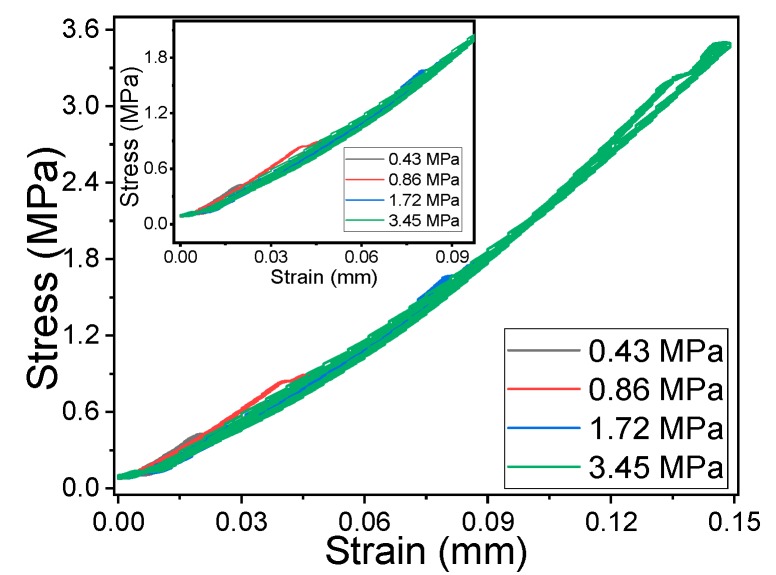
Stress-strain mechanical response for the 05CNT/PVDF composite, as representative for the rest of the nanocomposites.

**Figure 6 materials-12-03545-f006:**
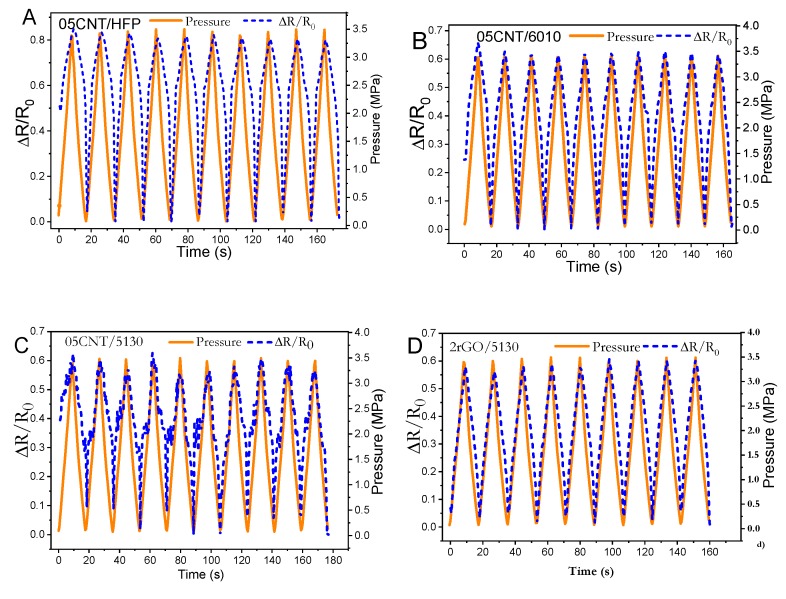
Electromechanical performance of the (**A**) PVDF-HFP, (**B**) 6010, and (**C**) 5130 with 0.5 wt.% of CNT for 10 cycles from unloading to 3.5 MPa of pressure. (**D**) PVDF 5130 reinforced with 2 wt.% CNT in cycles up to 3.5 MPa.

**Figure 7 materials-12-03545-f007:**
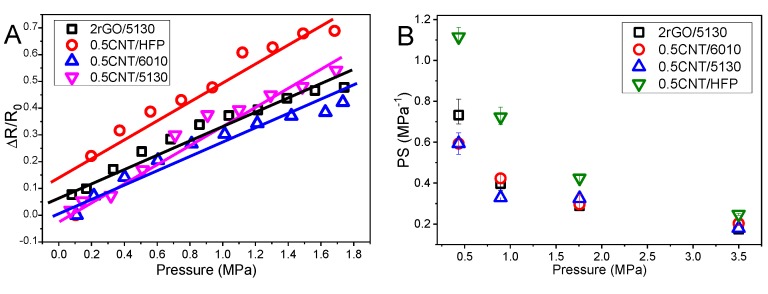
The electromechanical response of the different composites. Linearity between applied pressure and relative resistance variation in (**A**) and pressure sensitivity for composites with 0.5 wt.% of PVDF matrices and 2rGO/5130 composite (**B**).

**Figure 8 materials-12-03545-f008:**
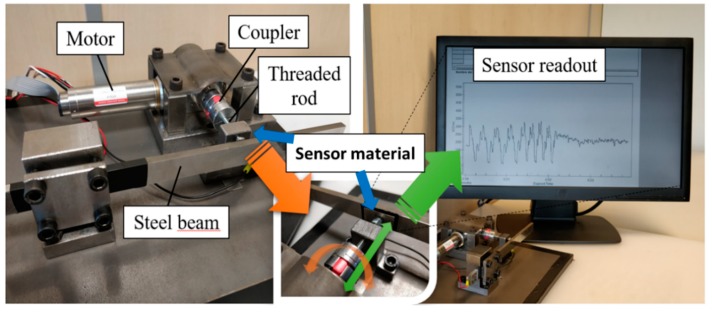
Fabricated test bench using CNT/PVDF as sensor material and obtained measurements.

**Table 1 materials-12-03545-t001:** Poly (vinylidene fluoride) (PVDF)-based polymers, nanofillers and solvent/polymer ratio used in the processing of the nanocomposites.

	DMPU/PVDF Vol/Vol	Carbon Nanotubes (wt.%)	Nomenclature	Reduced Graphene Oxide (wt.%)	Nomenclature
**PVDF 5130**	95/5	0 0.25 0.5 1	PVDF5130	0 0.5 1 2	PVDF5130
			0.25CNT/5130		0.5rGO/5130
			0.5CNT/5130		0.1rGO/5130
			1CNT/5130		2rGO/5130
**PVDF 6010**	90/10	0 0.25 0.5	PVDF6010		
			0.25CNT/6010		
			0.5CNT/6010		
**PVDF-Hexafluropropylene**	90/10	0 0.25 0.5	PVDF-HFP		
			0.25CNT/HFP		
			0.5CNT/HFP		

**Table 2 materials-12-03545-t002:** Mechanical parameters obtained from the stress-strain measurements for the different PVDF matrices and the corresponding composites with CNT and rGO.

Sample	Initial Modulus (MPa)	Strain at Rupture (%)	Stress at Rupture (MPa)	Yield Strain (%)	Yield Stress (MPa)
PVDF-HFP	356 ± 15	70.8 ± 4	15.5 ± 3	12.7 ± 3	18.1 ± 4
025CNT/HFP	372 ± 16	328.2 ± 16	17.4 ± 4	19.3 ± 5	19.1 ± 4
05CNT/HFP	439 ± 18	309.5 ± 15	22.4 ± 4	18.3 ± 4	22.3 ± 5
PVDF6010	1065 ± 45	11 ± 2	42.8 ± 8	9.6 ± 2	44.7 ± 10
025CNT/6010	1293 ± 49	11.1 ± 2	49.5 ± 10	8.9 ± 2	51.9 ± 11
05CNT/6010	1388 ± 51	13.6 ± 3	51.9 ± 10	8.3 ± 2	55.9 ± 12
PVDF5130	870 ± 40	23.4 ± 5	27.3 ± 6	9.8 ± 2	34.4 ± 7
025CNT/5130	863 ± 40	212.9 ± 13	30.0 ± 6	13.3 ± 3	42.0 ± 8
05CNT/5130	1244 ± 54	99.2 ± 5	32.0 ± 7	13.2 ± 3	43.6 ± 9
1CNT/5130	1220 ± 53	85.9 ± 4	38.6 ± 8	14.4 ± 4	46.6 ± 9
PVDF5130	870 ± 41	23.4 ± 4	27.3 ± 6	9.8 ± 2	34.4 ± 7
05rGO/5130	1151 ± 52	12.3 ± 3	39.6 ± 8	8.7 ± 2	43.2 ± 9
1rGO/5130	1327 ± 55	11.9 ± 3	53.4 ± 11	8.9 ± 2	56.1 ± 11
2rGO/5130	1265 ± 54	4.5 ± 1	44.1 ± 9	4.5 ± 1	44.1 ± 10

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
