# Peer review of "Electromechanical Properties of PVDF-Based Polymers Reinforced with Nanocarbonaceous Fillers for Pressure Sensing Applications"

_materials, 2019, doi:10.3390/ma12213545_

Round 1
Reviewer 1 Report
Two minor errors:
page 11, line 368 - a reference source not found,
page 12, line 386 - a reference source not found.
The paper sounds good and I recommend it for publication.
Author Response
Reviewer 1
Two minor errors:
page 11, line 368 - a reference source not found,
page 12, line 386 - a reference source not found.
Sorry for the mistakes. They have been corrected.
The paper sounds good and I recommend it for publication.
Thank for the positive evaluation of the manuscript and for supporting publication.

Reviewer 2 Report
In the manuscript, a study of PVDF-based materials for force compression sensing is presented. Different PVDF based polymers (PVDF-HFP, PVDF 6010 and PVDF 5130) and nanocarbonaceous fillers (carbon nanotubes and reduced graphene oxide) were used to tailor composites in view their mechanical, electrical and electromechanical properties.
Some comments are listed as follows.
The topic falls in the journal scope.
Moreover:
Aims of the paper should be better pointed out. Eqs. 2 - 3 - Samples of 30x10 mm of area and 20 to 60 μm of thickness are considered. Due to the small-scale, an analysis of the possibility of using standard formulas for stress and strain must be added.It is useful to check the existing reference on experimental analysis considering a small-scale parameter and a two-phase constitutive mixture. Reference to small-scale and nonlocal problems must then be added such as A nonlocal model with strain-based damage, International Journal of Solids and Structures, 46, 22-23, 2009, 4107-4122; On non-local and non-homogeneous elastic continua, International Journal of Solids and Structures, 46, 3-4, 2009, 651-676. Further references can be added to improve Introduction on stress driven models and other proposed models in literature. Section 3.5 – Are 8 cycles of 4 revolutions forward and back enough to confirm the suitability of the developed materials for sensor applications? English should be revised and misprints (such as “. 2.1Materials” or missing symbols “Error! Reference source not found”) should be checked.
Author Response
Reviewer 2
In the manuscript, a study of PVDF-based materials for force compression sensing is presented. Different PVDF based polymers (PVDF-HFP, PVDF 6010 and PVDF 5130) and nanocarbonaceous fillers (carbon nanotubes and reduced graphene oxide) were used to tailor composites in view their mechanical, electrical and electromechanical properties.
Some comments are listed as follows.
The topic falls in the journal scope.
Moreover:
Aims of the paper should be better pointed out. Eqs. 2 - 3 - Samples of 30x10 mm of area and 20 to 60 μm of thickness are considered. Due to the small-scale, an analysis of the possibility of using standard formulas for stress and strain must be added. It is useful to check the existing reference on experimental analysis considering a small-scale parameter and a two-phase constitutive mixture. Reference to small-scale and nonlocal problems must then be added such as A nonlocal model with strain-based damage, International Journal of Solids and Structures, 46, 22-23, 2009, 4107-4122; On non-local and non-homogeneous elastic continua, International Journal of Solids and Structures, 46, 3-4, 2009, 651-676. Further references can be added to improve Introduction on stress driven models and other proposed models in literature.
A: Mechanical properties of the PC, PLA and composites were evaluated up to rupture and cycles up to 3.5 MPa (maximum pressure for the specific application).
We appreciate the comment and suggestion, but the experimental measurements are the typical performed in these samples in the literature (see references) and it is very important that the results can be directly compared to them. Further, those standard formulas are again the ones used typically in the literature (see references). That the results are properly represented by the formulas is also demonstrated by the possibility of developing a functional application, that takes into consideration the values provided by the characterization of the materials. . Of course, the recommendations from the referee are interesting in order to get further insights on the developed materials and will be taken into account for further developments. IN order to make this position transparent we have included the following sentence:
The samples were considered as macroscopically homogeneous materials, without consider non-local effects [37] due the low filler content.
Section 3.5 – Are 8 cycles of 4 revolutions forward and back enough to confirm the suitability of the developed materials for sensor applications?
A: The performed measurements of the piezoresistive properties of the sensor is much larger than showed in Figure (8 cycles). The provided Figure as well the Supplementary Video shows one measurement of the several tests performed, all leading to reproducible results. In other way, the piezoresistive performance of the sensors is stable with cycles number, as can be observed in Figure 6 and in all experimental measurement of the sensor performance.
Figure 8 shows the sensor readout when submitted to 8 cycles of 4 revolutions forward and back, as representative performance of the piezoresistive sensor. The angular velocity was 750 rpm for the first 4 cycles and 1500 rpm for the latter four cycles.
English should be revised and misprints (such as “. 2.1Materials” or missing symbols “Error! Reference source not found”) should be checked.
A: All manuscript was revised to correct misprints and missing symbols.
Reviewer 3 Report
The manuscript "Electromechanical properties of PVDF-based polymers reinforced with nanocarbonaceous fillers for pressure sensing applications" compares PVdF fillers based on rGO and CNT in their pressure sensory ability. So far that topic is not real novel as many research been made for such material but their results have some novelty.
The manuscript needs some revision especially in their goal of work, the comparison to literature why their work better than others. As example the performance of PVDF 5130/0.5% CNT in Figure 6c shows irregularity in the deltaR/R0. Where comes those irregularity please give explanation. In Figure 7 there only a description of A but B is missing. Please correct that.
It would be also nice to show a real case where the sensor works not only the measurement device at Figure 8. Please include a sample test with read out of sensibility. There some errors of literature saying "Error! Reference source not found". Please correct that.
Author Response
Reviewer 3
The manuscript "Electromechanical properties of PVDF-based polymers reinforced with nanocarbonaceous fillers for pressure sensing applications" compares PVdF fillers based on rGO and CNT in their pressure sensory ability. So far that topic is not real novel as many research been made for such material but their results have some novelty.
A: We have developed composite materials for sensor applications. Processing, filler content and the mechanical and electromechanical properties are tailored to improve the piezoresistive sensibility of the materials.
The manuscript needs some revision especially in their goal of work, the comparison to literature why their work better than others. As example the performance of PVDF 5130/0.5% CNT in Figure 6c shows irregularity in the deltaR/R0. Where comes those irregularity please give explanation. In Figure 7 there only a description of A but B is missing. Please correct that.
A: Thanks for the suggestions. Some references were added in the introduction and discussion with respect to the literature in the field. The novelty is presented in the manuscript and is based in the use of polymer matrices with different initial modulus as well as different fillers, filler content and a real application. The irregularities are most provably due to the different wettability between filler and polymer, leading to interfaces that promote electromechancical noise, when compared to others filler/polymers. Figure 7 was corrected.
In this work, a comprehensive study of PVDF-based materials for force compression sensing is presented with excellent performance and linearity [37, 38].
Similar electromechancial linearity and cycling performance has been reported in [53, 54] with different materials and experimental approaches.
It would be also nice to show a real case where the sensor works not only the measurement device at Figure 8. Please include a sample test with read out of sensibility. There some errors of literature saying "Error! Reference source not found". Please correct that.
A: The have been corrected in the manuscript.
The real case is presented in the Supplementary video. Figure 8 is just an image of the sensor working. The sensor sensibility is characterized in Figure 7B. The real time electrical resistance measurements are carried out by:
“The resistance change produced under the compression pressure was measured between the fixtures of the beam and the rod employing a Fluke 8845A multimeter and logged through PC. The PMDC motor (Maxon EC-4 pole 22 mm) was actuated by an EPOS2 controller.”
Reviewer 4 Report
interesting study
well written and well reported
the final proof of concept provides added value
the manuscript can be accepted
Author Response
Thank for the positive evaluation of the manuscript and for supporting publication.
Round 2
Reviewer 2 Report
it can be published as it is.
Author Response
No comments
Reviewer 3 Report
All questions sufficient answered. manuscript can be accepted
Author Response
Thank you for supporting the manuscript